# Preparation of Drug-Loaded Liposomes with Multi-Inlet Vortex Mixers

**DOI:** 10.3390/pharmaceutics14061223

**Published:** 2022-06-09

**Authors:** Huangliang Zheng, Hai Tao, Jinzhao Wan, Kei Yan Lee, Zhanying Zheng, Sharon Shui Yee Leung

**Affiliations:** 1School of Pharmacy, The Chinese University of Hong Kong, Shatin, Hong Kong; huangliangzheng@cuhk.edu.hk (H.Z.); 1155131187@link.cuhk.edu.hk (K.Y.L.); 2Center for Turbulence Control, Harbin Institute of Technology, Shenzhen 518055, China; taohai1998@163.com (H.T.); 210340108@stu.hit.edu.cn (J.W.)

**Keywords:** MIVM, vortex mixing, micromixing, liposomes, computational fluid dynamics

## Abstract

The multi-inlet vortex mixer (MIVM) has emerged as a novel bottom-up technology for solid nanoparticle preparation. However, its performance in liposome preparation remains unknown. Here, two key process parameters (aqueous/organic flow rate ratio (FRR) and total flow rate (TFR)) of MIVM were investigated for liposome preparation. For this study, two model drugs (lysozyme and erythromycin) were chosen for liposome encapsulation as the representative hydrophilic and hydrophobic drugs, respectively. In addition, two modified MIVMs, one with herringbone-patterned straight inlets and one with zigzag inlets, were designed to further improve the mixing efficiency, aiming to achieve better drug encapsulation. Data showed that FRR played an important role in liposome size control, and a size of <200 nm was achieved by FRR higher than 3:1. Moreover, increasing TFR (from 1 to 100 mL/min) could further decrease the size at a given FRR. However, similar regularities in controlling the encapsulation efficiency (EE%) were only noted in erythromycin-loaded liposomes. Modified MIVMs improved the EE% of lysozyme-loaded liposomes by 2~3 times at TFR = 40 mL/min and FRR = 3:1, which was consistent with computational fluid dynamics simulations. In summary, the good performance of MIVM in the control of particle size and EE% makes it a promising tool for liposome preparation, especially for hydrophobic drug loading, at flexible production scales.

## 1. Introduction

Liposomes are the most studied and well-characterized nanocarrier for drug delivery. They are sphere-shaped vesicles comprising an aqueous core, surrounded by one or more phospholipid bilayers. They are characterized by their physical properties, including size, charge, lipid composition, number of lamellae, and surface modifications, which collectively govern their stability and in vivo behavior [1]. Liposomes have been used to carry drugs or vaccines on their surface via electrostatic interactions, and have been incorporated into the lipid bilayer for lipophilic drugs, or encapsulated into the aqueous core for hydrophilic compounds [2]. Liposomal formulations have provided a number of advantages over plain drug solutions, including reduced toxicity of encapsulated drugs [3], prolonged systemic circulation [4], controllable drug release kinetics [5], and site targeting [6]. These unique properties of liposomes to encapsulate both lipophilic and hydrophilic compounds allows them to deliver a diverse range of therapeutics, from small molecules to biologicals (nucleic acids (RNA and DNA) [7], peptides [8], proteins [9], and bacteriophage [10]). The first liposomal drug (AmBisome^®^) was approved in 1991 by the European Medicines Agency. The development of liposomal drug delivery systems has boomed since then, with more than 18 clinically approved liposomal formulations for the treatment of different diseases in clinical application to date [11].

The formation of liposomes is a self-assembly process driven by the hydrophobic interaction of phospholipid molecules in an aqueous phase. The particle size, encapsulation efficiency, and stability are mainly determined by the choice of the phospholipids, the drugs themselves, and the preparation methods [12]. The common methods used for liposome preparation can be understood as two steps: preparing the initial emulsion with vesicles which are typically heterogenous in size and multi-lamellar (thin-film hydration, reverse phase evaporation, ethanol/ether injection), followed by energy input (sonication, homogenization, extrusion) to obtain uniform, nano-size, and unilamellar liposomes [12]. Some of these liposome production methods impose harsh or extreme conditions, which can result in the denaturation of the phospholipid raw material and encapsulated drugs, particularly for the biological species. Furthermore, the traditional techniques are usually not readily scalable from the bench-scale to the industrial production. In this context, the feasibility of using a microfluidic approach to produce liposomes has attracted significant attention in the past decade [13]. A detail comparison between microfluidics and traditional extrusion for liposome production was performed by Shah et al. [14], where data showed both microfluidics and extrusion produce liposomes with identical in vitro and in vivo properties. A practical case of large batch liposome production with microfluidics from Web et al. [15] indicated that the microfluidics embraced good batch reproducibility for liposome production. These studies confirmed that microfluidic technology has been a hit in liposomes manufacturing.

A four-stream multi-inlet vortex mixer (MIVMs) has been previously demonstrated to be an excellent tool for the scale production of different polymeric drug nanoparticles, drug-alone nanoparticles, and solid lipid nanoparticles, based on the principle of flash nanoprecipitation [16,17,18,19]. Nanoparticles with good size uniformity and reproducibility can be achieved [20,21]. The unique design of MIVM enables an efficient mixing which is completed in milliseconds through vortex force. Moreover, the four inlet streams can independently contribute to the micromixing in the vortex chamber, allowing for greater flexibility in the nano-formulation process [17,19]. The feasibility of MIVM in preparing liposome vesicles has been proposed in a patent application [22], but no systematic investigation has been reported to date. Recently, Bokare et al. [23] reported that altering the inlet geometry from straight channels to herringbone-patterned channels could significantly increase the vorticity inside the mixing chamber, resulting in more uniform mixing, and hence yielded smaller lipid–polymer hybrid nanoparticles with more uniform size distribution. Previous studies have shown that the chaotic advection could be introduced to a single-phase flow in the tortuous channel at a sufficiently high Reynolds level (but still in the laminar region) to improve mixing [24,25]. We hypothesize that the change of inlet flow field by a tortuous channel would further improve the vortex mixing in the mixing chamber of the MIVM. In the present study, we first systemically investigated the applicability of MIVM in preparing drug-encapsulated liposomes via passive loading, which is widely adopted for the loading of emerging biological therapeutics. The impacts of the inlet geometries, straight channels, herringbone-patterned straight channels, and zigzag channels on the size, polydispersity, and encapsulation efficiency were also evaluated.

## 2. Materials and Methods

### 2.1. Materials

Soybean phosphatidylcholine (PC) with over 98% purity was obtained from Tywei Pharmaceutical Co., Ltd. (Shanghai, China). Cholesterol and phosphate-buffered saline (PBS) were purchased from Sigma–Aldrich (St. Louis, MO, USA). Lysozyme and erythromycin were obtained from Macklin Biochemical Co., Ltd. (Shanghai, China).

### 2.2. MIVM Inlet Designs

A conventional MIVM with straight inlet channels (MIVM-straight), as detailed in Liu et al. [17], and two MIVM with modified inlet geometries (one with herringbone-patterned straight inlet channels (MIVM-herringbone), and one with zigzag inlet channels (MIVM-zigzag)) were used in the present study (Figure 1). The critical design parameters of the MIVMs are shown in Figure 1A. The MIVM-herringbone had repeated patterns of grooves on the bottom of the straight inlet channels, with detailed dimensions depicted in Figure 1B. The MIVM-zigzag replaced the straight inlet channels with zigzag paths, with a channel width of 0.42 mm (Figure 1C). All MIVM mixers were fabricated with stainless steel by computer numerical control machining with a surface roughness of 3.2 µm Ra (Zhuanxin Precision, Guangdong, China). The MIVM was connected to the syringe pumps (Model PHD 2000, Harvard Apparatus, MA, USA), according to Figure 1D, with two opposite inlet streams for the organic phase and another two for the aqueous phase.

### 2.3. Preparation of Blank and Drug-Loaded Liposomes with the Conventional MIVM

For blank liposome preparation, the aqueous phase was PBS, and the organic phase contained various amounts of PC and cholesterol (at a fix molar ratio of 3:2) dissolved in ethanol to study the effects of lipid concentrations. To evaluate the influences of flow parameters on the size of the liposomes at a selected lipid concentration (40 mg/mL), the aqueous and organic phases were injected into the conventional MIVM over a wide range of total flow rate (TFR, 1–120 mL/min) and aqueous to organic flow rate ratio (FRR, 1:1–19:1). The formed liposomes were collected at the outlet of the MIVM with a connecting PTFE tube (internal diameter = 1.80 mm and length = 340 mm). The lysozyme- or erythromycin-loaded liposomes were prepared with either MIVM-straight, MIVM-herringbone, or MIVM-zigzag models. As a model hydrophilic compound, lysozyme was dissolved in the aqueous phase at a concentration of 20 mg/mL. The hydrophobic erythromycin was dissolved in the organic phase at a concentration of 20 mg/mL. In order to obtain representative liposome samples, the MIVM was kept running smoothly for at least 20 s, and at least 10 mL for all liposome batches was collected.

### 2.4. Analysis of Particle Size and Particle Size Distribution

The size and particle size distribution of the produced liposomes were analyzed via dynamic light scattering (DLS) using a Delsa^TM^ Nano C analyzer (Beckman Coulter, Inc., Brea, CA, USA). The equipment was coupled with a dual 30 mW laser that emits at 658 nm with a scattering angle of 165°. The intensity–mean diameter (Z-average) and the polydispersity index (pdi) of the liposome formulations prepared from different flow conditions were obtained. All measurements were performed in triplicate at 25 °C.

### 2.5. Transmission Electron Microscopy (TEM)

The morphology of the produced liposomes was observed with TEM. TEM samples were prepared on carbon coated copper grids 200 mesh (FCF300-CU) supplied from Electron Microscopy Sciences (Philadelphia, PA, USA). A drop of liposome sample was placed onto the TEM grid for 5 min, and excess of the sample was blotted. The samples were then negatively stained with 1% phosphotungstic acid (PTA) adjusted to pH 7.4 with 0.1 N sodium hydroxide. Any excess stain was blotted, and the grids were left to air-dry. Observations were made using FEI Tecnai^TM^ G^2^ Spirit Twin (Hillsboro, OR, USA) at 120 keV and ×4.2 k magnification.

### 2.6. Encapsulation Efficiency Determination

For the liposomal lysozyme, concentrations of unentrapped lysozyme and total lysozyme were quantified via bicinchoninic acid (BCA) protein assay. Briefly, 20 μL liposomal lysozyme was added and reacted directly with 180 μL BCA reagent in the 96-well plate to quantify the extra-liposomal lysozyme (C_extra_). Another 20 μL liposomal lysozyme was reacted with 180 μL BCA reagent after being lysed by 0.1% Triton X-100 to quantify the total lysozyme (C_total_). The encapsulation efficiency (EE%) was calculated as EE% = (C_total_ − C_extra_)/C_total_ × 100%.

For the liposomal erythromycin, the unentrapped erythromycin was removed by a 0.4 μm millipore filter. Then, the filtered liposomal erythromycin (C_intra_) and unfiltered liposomal erythromycin (C_total_) were dissolved by methanol to five times the volume, and quantified by Agilent 1290 HPLC system with the Agilent C18 column (250 mm × 4.6 mm, 5 µm), detected at 215 nm. The mobile phase was acetonitrile: methanol: water 50:10:40 (*v*/*v*/*v*), at a flow rate of 1 mL/min at 40 °C. The encapsulation efficiency (EE%) was calculated as EE% = C_intra_/C_total_ × 100%. To further identify the drug/lipid molar ratio of liposomal erythromycin, the cholesterol (C_cholesterol_) in the liposomal erythromycin was detected using the same HPLC system and C18 column at 205 nm. The mobile phase was acetonitrile: methanol 60:40 (*v*/*v*), at a flow rate of 1 mL/min at 40 °C. The erythromycin/cholesterol molar ratio was calculated as Molar Ratio = (C_intra_/MW_erythromycin_)/(C_cholesterol_/MW_cholesterol_).

### 2.7. Modeling of Fluid Mixing

The fluid mixing behavior between ethanol and water within the MIVM devices was studied numerically using computational fluid dynamics (CFD) simulations. The flow was assumed to be in a steady, laminar regime, and the corresponding governing equations, including mass, momentum, and advection–diffusion equations, were solved using the finite volume CFD code ANSYS CFX 17.2, where the advection–diffusion equation was configured by defining an additional variable of ethanol concentration, and the mixture density was set to be a function of the ethanol concentration. It should be noted that a transition from steady to transient flow is possible for non-straight devices, as TFR is sufficiently high (approximately 100 mL/min based on a previous study using a geometry resembling the MIVM-zigzag channel [26]). Nevertheless, the transient effect is only significant at locations far from the inlet (approximately 10 repeating zigzag patterns based on [26]), hence the steady-state assumption should at least indicatively capture the flow mixing of different devices. For each phase, its fluid properties, including density and dynamic viscosity, were assumed to be constant, and the diffusion coefficient of ethanol in water was set to 0.8 × 10^−9^ m^2^/s [27].

A scheme of the computational domain and details of surface mesh is shown in Figure 2. For this, two aqueous phase and two organic phase inlets were arranged alternatively; fluid mixture exited the computational domain from a single outlet. Unstructured tetrahedral mesh was used for the entire domain, and the grid was refined within the mixing chamber of the MIVM device, especially in regions where the flow did not develop smoothly. A total number of 1.52 × 10^−6^ mesh elements were included to guarantee mesh-independent solutions. Constant velocities were set at different inlets to ensure the same organic/aqueous flow rates as in experiments; a pressure boundary condition was applied at the outlet and at the wall, and the usual no-slip condition was used. In the simulations, a modified Rhie–Chow algorithm was used to link the pressure and velocity fields, which were solved via a coupled solver. A second-order-bounded differencing scheme was used for the convective terms. The system was regarded as having reached a converged state once all scaled residuals fell below 10^−6^ and the global imbalances, representing overall conservation, fell below 10^−3^.

### 2.8. Statistical Analysis

All results were expressed as mean ± one standard deviation, unless specified otherwise. Two-way analysis of variance (ANOVA), along with Turkey’s multiple comparisons, was performed to compare the effect of two factors (TFR and FRR; TFR and different MIVMs) under multiple levels in particle size and encapsulation efficiency of liposomes. A *p* value of <0.05 was considered as statistically significant.

## 3. Results and Discussion

### 3.1. Preparation of Blank Liposomes with Conventional MIVM

The microfluidic-assisted liposome formation is achieved via a solvent exchange mechanism, for which lipid bi-layered fragments first form at the aqueous–organic solvent interface, followed by self-assembly into liposomes upon increased polarity of the surrounding medium [28,29]. The whole process occurs within milliseconds, and is affected by the local lipid concentration, the volumetric ratio between aqueous and organic phases, and the mixing efficiency. Therefore, the applicability of the conventional MIVM in preparation liposome formulations was first assessed across a wide range of lipid concentrations, aqueous to organic phases FRR and TFR, which are altogether affecting the aforementioned parameters that govern the liposome formation process.

#### 3.1.1. Effect of Lipid Concentration

The effect of lipid concentration (ranged from 8–80 mg/mL) in the organic phase on the size of liposomes prepared by the conventional MIVM was first assessed. Overall, liposomes could be produced with an acceptable size (≤200 nm) and polydispersity (≤0.2) at suitable flow conditions. Representative size distribution profiles and TEM images of the produced liposomes are shown in Figure 3.

According to Table 1, liposomes prepared under the same flow conditions had comparable average size, regardless of the lipid concentrations. The PDI values of all of the produced liposomes varied between 0.15 to 0.25, and no specific trends were noted with respect to the lipid concentration, FRR, and TFR. These observations were different from liposomes prepared with other microfluidic systems, in that the size of the liposomes and their dispersity generally increased with increasing lipid contents [13,30,31,32]. This property of MIVM could allow flexible production scale-up by simply varying the lipid concentrations without changing the flow conditions and the configuration of the set-up, which may prove superior over other microfluidic setups. In the follow-up experiments, we chose a lipid concentration of 40 mg/mL in the organic phase for study, as this would yield satisfactory liposome concentrations for large-scale production.

#### 3.1.2. Effect of FRR

At the same total flow rate, Figure 4 clearly shows that the size of liposomes decreased with increasing aqueous to organic FRR ratios. The reduction was more profound for FRR between 1:1 to 5:1, and levelled off with a size of ~160 nm for higher FRR values. Such a trend was similar to liposomes prepared with different microfluidic devices and configurations [13,32,33], but for the FRR to reach the plateau value in liposome size with these setups, the values required were much higher, typically beyond 20:1. The disparity is likely attributed to the different mixing mechanisms between the flow-focusing microfluidic setups and the MVIM; the mixing of aqueous and organic solvents in the former is dominated by diffusion, while the latter is mainly achieved by chaotic micromixing [17]. Lee et al. [34] showed experimentally that the aqueous/organic interface increased with increased FRR, thereby increasing the diffusive mixing of solvents to form smaller liposomes. The improved mixing environment within the MIVM could rapidly create local saturated lipid zones for liposome formation, leading to the achievement of the minimum size at a lower FRR. This property would also be favorable for liposome production as a higher final lipid concentration can be obtained, representing a greater flexibility in scaling up the production yield of liposomes with the MIVM.

#### 3.1.3. Effect of TFR

Figure 5 shows that the size of liposomes generally decreased with increasing TFR for a given FRR (1:1 and 3:1). In most previous studies on microfluidic-assisted liposome formation, the liposome size was found to be insensitive to or only weakly-dependent on changes in TFR [10,13,33,35]. As the length scale of the microfluidic devices increased to ≥1 mm, a stronger inverse correlation between the size and TFR was reported [13,30,34]. Yanar et al. suggested that this is because the mixing mechanism would change from diffusion-dominant to advection-dominant as the channel dimension increased, and such an effect was more prominent in curved channels due to the generation of secondary flows. They also showed computationally that the mixing efficiency increased with increasing TFR (and FRR), promoting the rapid assembling of lipid bilayer fragments into smaller liposomes. It is noteworthy to mention that the TFR can be achieved at higher concentrations than 100 mL/min, which are much higher than most microfluidic devices investigated for liposome formation, which generally operate at TFR < 1 mL/min, offering a distinct advantage for mass production.

### 3.2. Preparation of Drug-Loaded Liposomes with Conventional MIVM

In general, drug incorporation into liposomes can be achieved either passively upon the liposome formation process or actively after the liposome formation [12]. Hydrophilic (water-soluble) drugs are loaded inside the aqueous core of liposomes, while hydrophobic drugs are incorporated into the lipid bilayer of liposomes. Passive encapsulation of hydrophilic drugs depends on the ability of liposomes to trap the dissolved drugs along with the aqueous buffer during liposome formation. In other words, the drug encapsulation effectiveness is largely limited by the trapped volume within the liposomes and the drug solubility. In general, the encapsulation efficiency of passively loaded hydrophilic small molecules is around 10–30%. For hydrophobic drugs, the amount of drug incorporation is governed by drug–-lipid interactions and the drug solubility within the lipid bilayer. Encapsulation efficiency of 100% is sometimes achievable. Here, we chose lysozyme and erythromycin as model hydrophilic and hydrophobic drugs, respectively, to investigate the feasibility of MIVM in entrapping drugs into liposomes via a passive loading approach.

#### 3.2.1. Lysozyme-Loaded Liposomes

Similar to the blank liposomes, the size of the lysozyme-loaded liposomes gradually decreased with increasing TFR, from ~200 nm at 1 mL/min to ~110 nm at 100 mL/min (Figure 6A). Statistically significant differences were noted between the three studied FRR (3:1, 9:1 and 19:1) at a given TFR, but the actual difference was mostly minor. In addition, lysozyme-loaded liposomes had a comparable size to the blank liposome, suggesting that the loading of drugs within the aqueous cores did not cause significant alteration of the liposome formation process. Figure 6B shows the encapsulation efficiency of the lysozyme into the liposomes. The encapsulation efficiency was around 30% for most conditions studied. As previous discussion, the encapsulation efficiency would depend on the aqueous volume trapped within the liposomes. For the same starting lipid concentration, the volume fraction of the aqueous phase to be trapped inside the aqueous cored decreased as the FRR increased, with a decrease in the encapsulation efficiency expected. Furthermore, the larger specific surface area caused by the smaller particle size with increasing TFR would further decrease the encapsulation efficiency when using the same mass of lipids. Surprisingly, the encapsulation efficiency for higher FRR (9:1 and 19:1) was slightly higher than that obtained with FRR = 3:1, and no significant differences were noted for the TFR range studied at a given FRR. For protein encapsulation, the interaction between proteins and lipids could play a critical role in determining the encapsulation efficiency, and higher protein encapsulation efficiency could be achieved beyond the volume fraction of the aqueous core [36]. With optimal lipid composition and process parameters, an encapsulation efficiency above 40% has been reported previously [37,38], suggesting that the promotion of interaction between lipid bilayer fragments and lysozyme can improve encapsulation efficiency. Therefore, the similar (or slightly improved) encapsulation efficiency at a higher FRR could possibly be attributed to a better mixing efficiency at a higher FRR.

#### 3.2.2. Erythromycin-Loaded Liposomes

Figure 7A shows the size of erythromycin-loaded liposomes at various FRR and TFR. It was noted that the size of liposomes decreased in size with increasing TFR, following the same trend as the blank liposomes and lysozyme-loaded liposomes. Generally, there was no difference in the size for erythromycin-loaded liposomes prepared from the three studied FRR, with only a few exceptions at FRR = 19:1. As shown in Figure 7B, there was an obvious improvement in the encapsulation efficiency with gradually increasing FRR from 3:1 to 19:1. Different from the drug loading process of the hydrophilic lysozyme, the loading of the hydrophobic erythromycin depends on reversed phase precipitation. A higher FRR could cause a higher instant ethanol diffusion gradient, followed by faster precipitation of the hydrophobic erythromycin. Thus, it could be inferred that more crystalline or amorphous erythromycin precipitates can be incorporated into the lipid bilayer of liposomes upon the liposome self-assembly process at a higher FRR [39,40]. Under the fixed FRR, however, there was no significant variation in the EE% of erythromycin liposomes with the increase of TFR, where the FRR 3:1 was 25%, the FRR 9:1 was 40%, and the FRR 19:1 was 50%, respectively. Since the loading of hydrophobic erythromycin was directly related to the lipid content, the drug/lipid molar ratio of erythromycin-loaded liposomes was also evaluated. No significant difference in the erythromycin/cholesterol molar ratio was noted with increasing TFR at a given FRR (Appendix A). On the other hand, a significantly higher erythromycin/cholesterol molar ratio was detected at higher FRRs, similar to the trend noted in EE%. These results suggested that a higher drug/lipid molar ratio could be achieved by a higher FRR because the processing conditions are not likely to cause phospholipid loss or degradation. Overall, only FRR was identified as the critical process parameter (CPP) able to achieve a tunable encapsulation efficiency and drug/lipid molar ratio for erythromycin liposomes, which can guide us to understand the unpredictable change of encapsulation efficiency during scale-up or production line transformation. Compared with the lysozyme-loaded liposomes with unclear CCPs to EE%, it can be concluded that the MIVM showed better control of drug encapsulation in erythromycin-loaded liposomes via tuning FRR.

### 3.3. Effect of Inlet Geometries of MIVM

Up to here, we have demonstrated that the conventional MIVM (MIVM-straight) is suitable for the preparation blank and drug-loaded liposomes at flexible production scale with controllable size and polydispersity due to its excellent micromixing behavior [17]. Recently, Bokare et al. [23] reported an intriguing finding: altering the inlet geometry of the MIVM could further improve its mixing efficiency, yielding more lipid–polymer hybrid nanoparticles with a smaller size and better uniformity. Therefore, we next investigated whether this will also be noted for the production of liposomes with MIVM. One MIVM with the herringbone groove in the straight inlet channels, as reported in Bokare et al. [23], and one MIVM with zigzag inlet channels, reported to introduce chaotic mixing of single phase flow [24,25], were studied at a chosen FRR = 3:1, which would result in a high liposome yield and an acceptable liposomal size.

#### 3.3.1. Computationally Determined Mixing Efficiency

The mixing behavior of ethanol and water within the MIVM with different inlet geometries was characterized using CFD simulations. At a given FRR (3:1), the mixing became more effective with increasing TFR, irrespective of the inlet geometries of the MIVMs (Figure 8). For the same flow conditions, no apparent difference was noted at the lowest studied TFR (1 mL/min). As the TRF increased, the mixing of ethanol and water became more uniform. Figure 9A shows the mass fraction of ethanol (α_ETOH_) along a line across the mixing chamber. For MIVM-straight, none of the TFR could achieve uniform mixing within the chamber. On the other hand, the value of α_ETOH_ was closer to a uniform mixing ratio (0.25, indicated by the black dot line in Figure 9A) as the TFR increased to 40 mL/min and higher for the other two MIVMs. For a better comparison of the mixing performance of the three MIVM devices, both the absolute mixing index (AMI) and relative mixing index (RMI) were also calculated according to the simulation results. The former was defined as the ratio of the standard deviation of α_ETOH_ to mean α_ETOH_ over a specific location, and the latter was defined as the ratio of the standard deviation of α_ETOH_ over a specific location to the standard deviation of α_ETOH_ in the unmixed state, as indicated in the literature [41]. According to these definitions, the values of AMI close to 0 and RMI close to 1 indicate the state of perfect mixing. Figure 9B shows both AMI and RMI along a line across the mixing chamber. Both of them clearly indicate that the mixing efficiency increased with increasing TFR, and followed the order of MIVM-zigzag > MIVM-herringbone > MIVM-straight for all flow conditions, except at the very low TFR (1 mL/min).

#### 3.3.2. Size of Liposomes

For the conventional MIVM (MIVM-straight), it was noted that the liposome size decreased with increasing TFR (Figure 4 and Figure 10), accounted for by the improved mixing between the organic and aqueous phase (Figure 8 and Figure 9). Although this tendency was repeatable on two modified MIVMs with enhanced mixing efficiency aroused from the herringbone and zigzag inlet geometries, as noted in computation results, there were significant alterations of the size for the lysozyme-loaded and erythromycin-loaded liposomes (Figure 10). For the lysozyme-loaded liposomes, compared with MIVM-straight, MIVM-herringbone significantly increased the particle size at 40 mL/min, but reduced at 100 mL/min. The size of liposomes prepared by MIVM-zigzag decreased at 1 mL/min, but increased at 40 and 100 mL/min. For the erythromycin-loaded liposomes, compared with MIVM-straight, MIMV-herringbone reduced the size at 1 and 10 mL/min, but rebounded at 100 mL/min, and MIVM-zigzag reduced the size at 1, 40, and 100 mL/min. Thus, the laws of these variations caused by modified inlet geometry remained unclear and unpredictable. This observation somewhat different from that reported by Bokare et al. [23], that the MIVM with a herringbone inlet geometry could significantly reduce the size and the uniformity of lipid–polymer hybrid nanoparticles compared with the conventional MIVM. In fact, this was not completely surprising, as the nature of the nano-carriers are fundamentally different. Liposomes are vesicles, with their formation depending on the assembly and disassembly of the lipid bilayer fragments upon the mixing of the aqueous and organic phases. On the other hand, lipid–polymer hybrid nanoparticles with a solid core are formed by nucleation of the hydrophobic active compounds, followed by the aggregation of amphiphilic polymer to arrest further growth [16].

#### 3.3.3. Drug Encapsulation Efficiency

Comparing the data between three MIVMs, the enhanced mixing achieved with the modified inlet geometries generally failed to translate into higher encapsulation efficiency. No significant difference in the encapsulation efficiency of both lysozyme-/erythromycin-loaded liposomes was noted among the three MIVMs for most of the studied conditions, with a few exceptions (Figure 11). For the loading of lysozyme into the liposomes, it is interesting to note that, at a flow rate of 40 mL/min, the EE% was significantly augmented for the two modified MIVMs when compared with MIVM-straight (from ~15% to ~45%). Although this is a desirable outcome, the exact reason causing this subtle enhancement at this particular flow condition is unclear. Taking a closer look of the CFD results (Figure 9), the 40 mL/min condition did show a better mixing performance when compared with other flow rates, and the difference in the mixing efficiency was the largest between the modified MIVMs and MIVM-straight. For erythromycin-loaded liposomes, only using MIVM-zigzag with TFR 10 mL/min could significantly improve the encapsulation efficiency (from ~25% to ~35%), while other data showed no significant difference. In contrast, the drug/lipid ratio data in Appendix A indicated that only the 40 mL/min TFR in MIVM-zigzag could significantly improve the erythromycin/cholesterol molar ratio, which was consistent with CFD outcomes. These experimental results implied that the modified MIVMs could only improve the drug encapsulation under certain conditions, and varied with different drug compounds. For hydrophobic drug loading, the drug/lipid ratio seemed to be more relevant to the mixing efficiency than the encapsulation efficiency in the present work. However, the underlying governing parameters between the mixing efficiency and the liposome encapsulation efficiency/drug-lipid ratio remain unclear. Nevertheless, the good agreement between the measured data and CFD predicted results suggests that CFD simulation could be a useful tool in identifying favorable operating conditions. Moreover, a specially designed MIVM for special liposomal drugs could create higher technical barriers for original products, making them more competitive than generic products.

## 4. Conclusions

This study explored the feasibility of preparing liposomes via MIVM. The lipid concentration in the organic phase had no significant effect on the particle size of liposomes under the same process parameters. FRR was identified as the key factor for liposome size control, which facilitated the generation of liposomes with a size <200 nm when FRR was at least 3:1. TFR also played a role in the size control, in that smaller liposomal size can be obtained at higher TFR for both blank and drug-loaded liposomes. To load drugs passively into liposomes using a MIVM, controlling the encapsulation efficiency via tuning the flow parameters was achieved with the hydrophobic erythromycin loading, but not with the hydrophilic lysozyme. According to the CFD simulation, MIVM with modified inlet geometries (MIVM-Herringbone and MIVM-Zigzag) could indeed improve the mixing efficiency of the MIVM mixer. However, the enhanced mixing did not have significant impacts on the size or polydispersity of formed liposomes. Nonetheless, significantly higher EE% could be achieved by the modified MIVMs in lysozyme-loaded liposomes at a specific TFR (40 mL/min) for the studied FRR (3:1). Taken together, this study confirmed the applicability of MIVM for liposome preparation, and likely promotes its application both in liposome laboratory preparation and industrial manufacturing as a single-step liposome production technology.

## Figures and Tables

**Figure 1 pharmaceutics-14-01223-f001:**
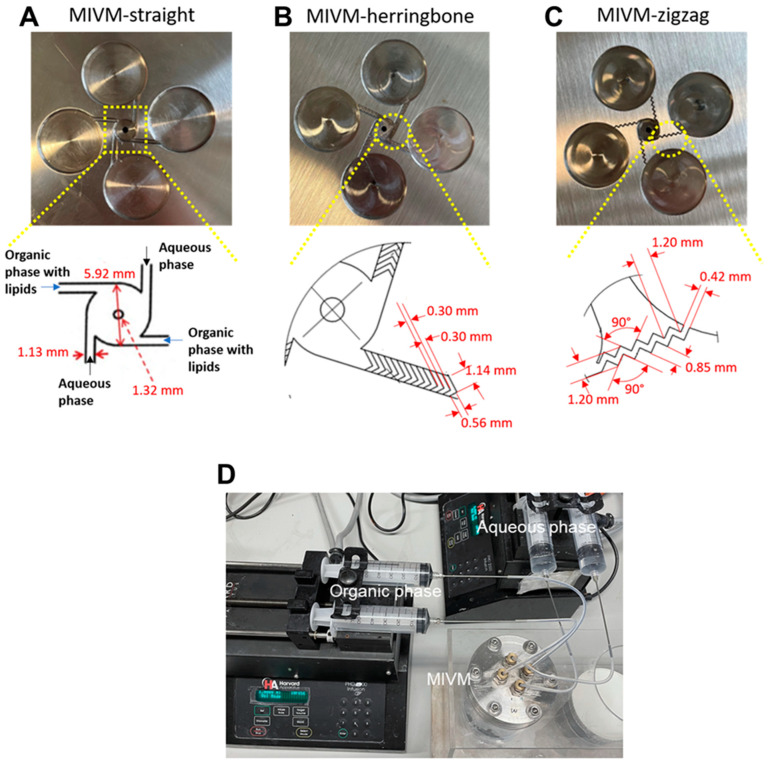
Image and critical parameters of the MIVMs used in the present study. (**A**) Conventional multi-inlet vortex mixer (MIVM-straight); (**B**) MIVM with modified herringbone-patterned inlets (MIVM- herringbone); and (**C**) MIVM with modified zigzag inlets (MIVM-zigzag). (**D**) Image of the experimental setup.

**Figure 2 pharmaceutics-14-01223-f002:**
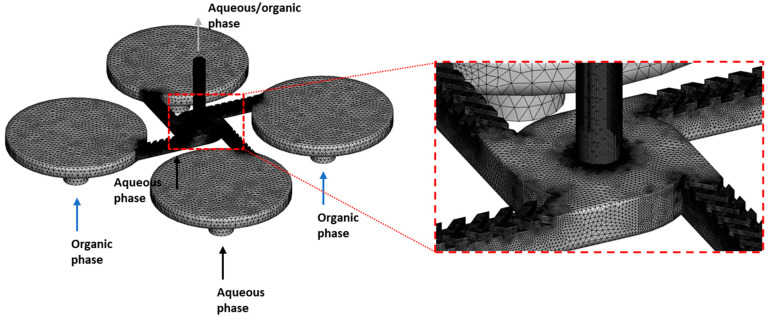
Scheme of the computational domain and details of surface mesh in CFD simulations.

**Figure 3 pharmaceutics-14-01223-f003:**
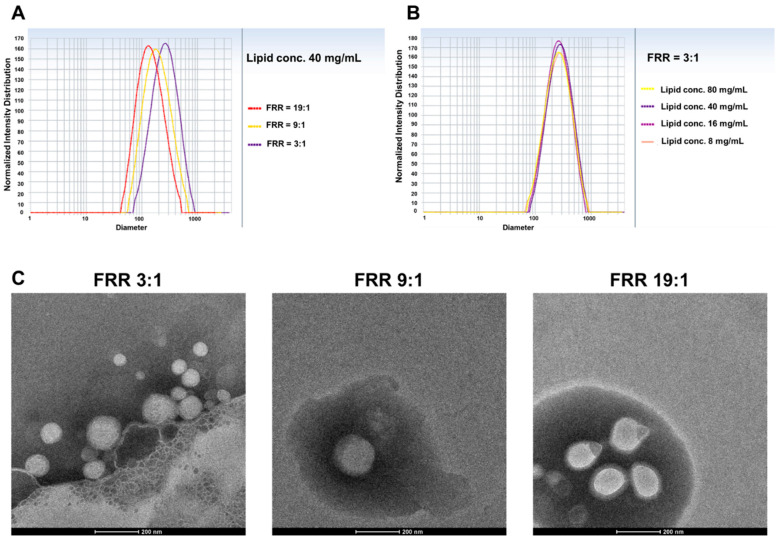
DLS representative size distribution profile of liposomes produced by the conventional MIVM with (**A**) fixed lipid concentration 40 mg/mL, and (**B**) fixed FRR 3:1. (**C**) The morphology and particle size of prepared liposomes under TEM with fixed lipid concentration at 40 mg/mL.

**Figure 4 pharmaceutics-14-01223-f004:**
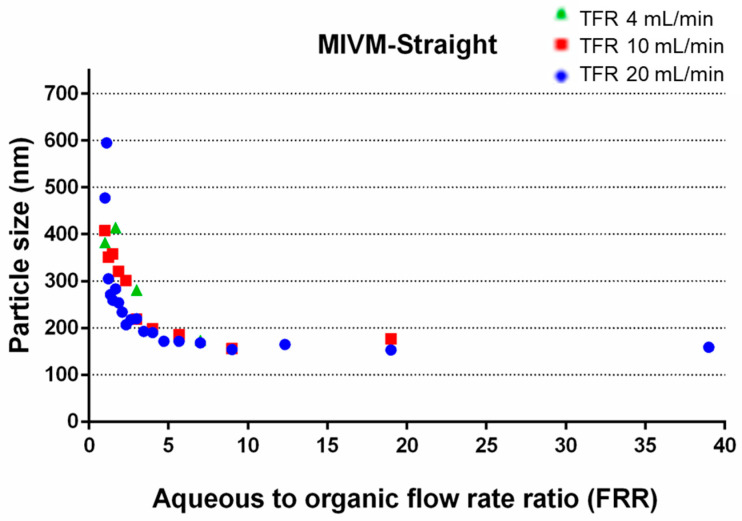
The effects of FRR on the z-average size of liposomes.

**Figure 5 pharmaceutics-14-01223-f005:**
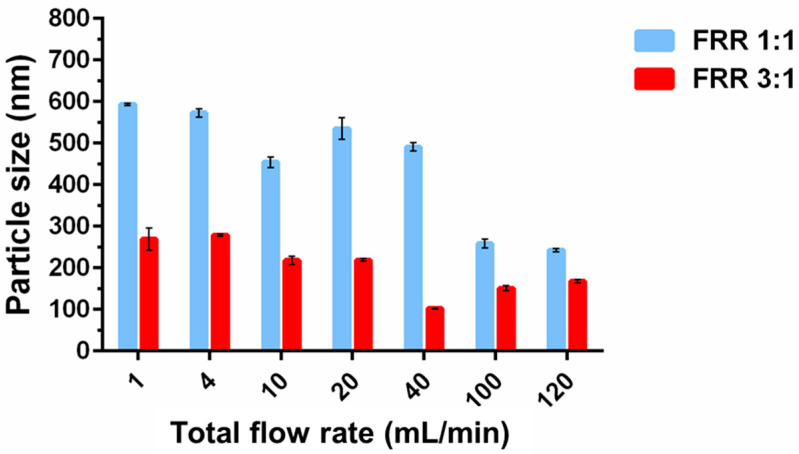
The particle size of liposomes prepared by flow rate under fixed relative flow rate: FFR = 1:1 and FFR = 3:1. Data presented as mean ± SD (*n* = 3).

**Figure 6 pharmaceutics-14-01223-f006:**
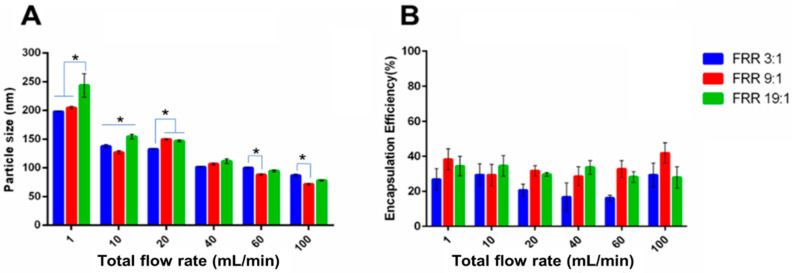
(**A**) The particle size and (**B**) the encapsulation efficiency of lysozyme-loaded liposomes prepared by MIVM-straight. Data presented as mean ± SD (*n* = 3) and *p* < 0.05 was marked * as statistically significant.

**Figure 7 pharmaceutics-14-01223-f007:**
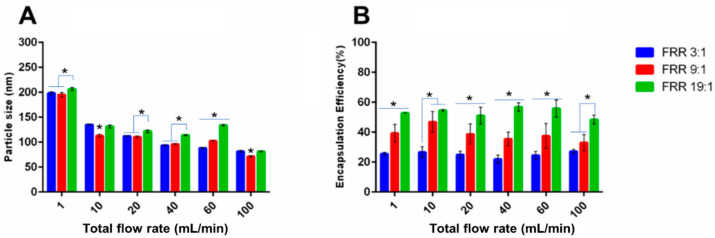
(**A**) The particle size and (**B**) the encapsulation efficiency of erythromycin-loaded liposomes prepared by MIVM-straight. Data presented as mean ± SD (*n* = 3), and *p* < 0.05 was marked * as statistically significant.

**Figure 8 pharmaceutics-14-01223-f008:**
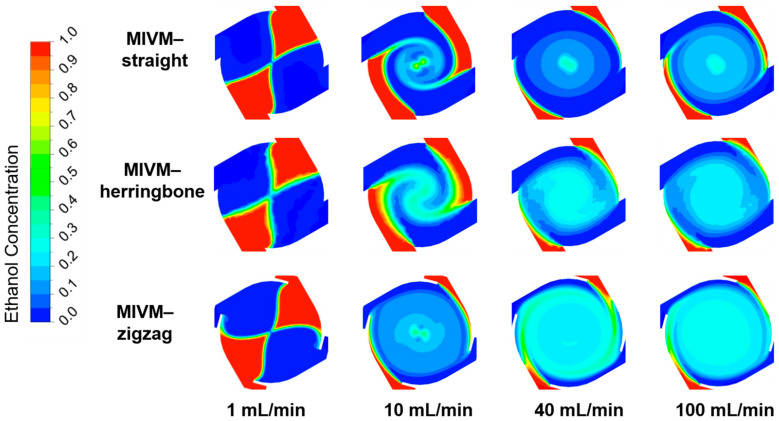
Contour plots of the ethanol mass fraction within the mixing chambers of the three MIVMs at various TFR with a given FRR (3:1).

**Figure 9 pharmaceutics-14-01223-f009:**
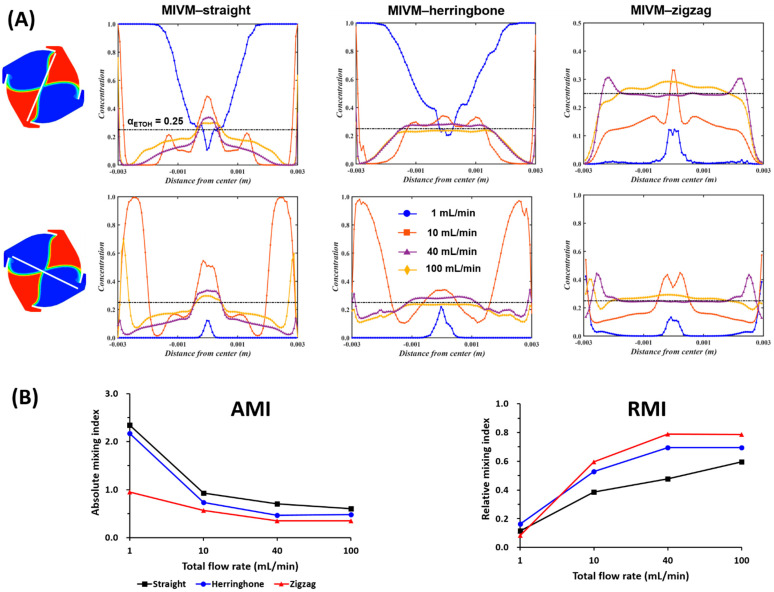
(**A**) Mass fraction of ethanol and (**B**) mixing index at a line across the mixing chamber (indicated by a white line on the left panel) at various TFR with a given FRR (3:1).

**Figure 10 pharmaceutics-14-01223-f010:**
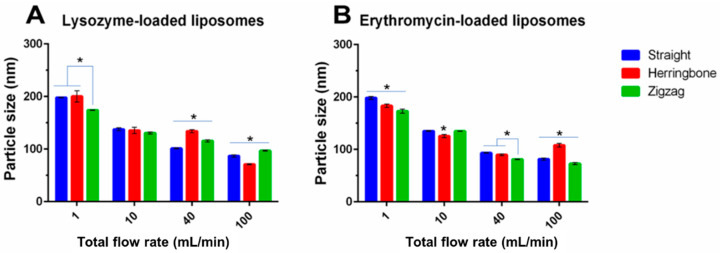
The size of (**A**) lysozyme-loaded and (**B**) erythromycin-loaded liposomes prepared with the three MIVM devices at various TFR with a given FRR (3:1). Data presented as mean ± SD (*n* = 3), and *p* < 0.05 was marked * as statistically significant.

**Figure 11 pharmaceutics-14-01223-f011:**
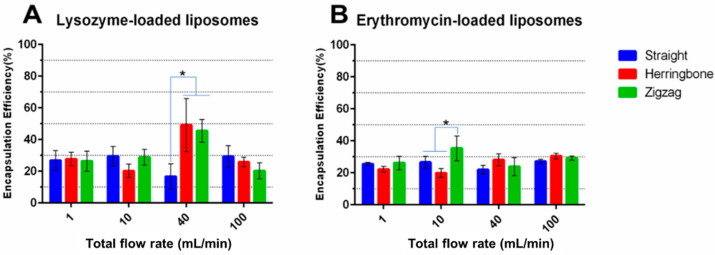
The encapsulation efficiency of (**A**) lysozyme-loaded and (**B**) erythromycin-loaded liposomes prepared with the three MIVM devices at various TFR with a given FRR (3:1). Data presented as mean ± SD (*n* = 3), and *p* < 0.05 was marked * as statistically significant.

**Table 1 pharmaceutics-14-01223-t001:** The particle size and PDI of liposomes prepared by MIVM-straight with various lipid concentrations and organic/aqueous flow rates.

Lipid Conc. (mg/mL) ^1^	FRR	TFR (mL/min)	Particle Size (nm) ^2^	pdi ^2^
8	3:1	4	231.5 ± 3.9	0.188 ± 0.016
16			247.7 ± 6.9	0.186 ± 0.029
40			227.5 ± 3.9	0.230 ± 0.022
80			248.9 ± 3.3	0.282 ± 0.019
8	9:1	10	144.7 ± 3.1	0.235 ± 0.037
16			185.8 ± 3.2	0.231 ± 0.018
40			167.0 ± 1.6	0.157 ± 0.028
80			172.3 ± 1.6	0.174 ± 0.020
8	19:1	20	120.9 ± 0.7	0.221 ± 0.009
16			126.5 ± 3.7	0.245 ± 0.013
40			124.2 ± 0.7	0.219 ± 0.017
80			138.3 ± 2.5	0.202 ± 0.023

^1^ conc. = concentration; ^2^ Value presented as mean ± SD.

## Data Availability

Not applicable.

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
