# Peer review of "Preparation of Drug-Loaded Liposomes with Multi-Inlet Vortex Mixers"

_pharmaceutics, 2022, doi:10.3390/pharmaceutics14061223_

Round 1

Reviewer 1 Report

The manuscript is interesting since it presents a new way for the production of unloaded and loaded liposomes using multi-inlet vortex mixer. The experimetal work is comprehensive and reasonable.

I suggest to better introduce the multi-inlet vortex mixer techniques (MIVM) and flash precipitation in the manuscript for a better readability. Moreover, despite it has been stated that this technique has been already employed for the production of nanoparticles, the previous literature has not sufficiently reported and discussed. A paragraph reporting the comparison between MIVM and other techniques for the production of liposomes can be also useful to improve the manuscript.

Reviewer 2 Report

The manuscript entitled “Preparation of drug-loaded liposomes with multi-inlet vortex mixers” describes a botton-up technique for preparation of drug-loaded liposomes and analyzes the factors affecting size and entrapment efficiency of the formed liposomes. This is an interesting work providing information about  a promising tool for liposome preparation at laboratory and industrial scale.

Points to be addressed:

-          Pag 3 line 105: “… various amounts of PC and colesterol (at a fix weigth ratio of 3:1)…”

Molar ratio is more relevant than weight ratio for liposome bilayer composition. Please provide PC/colesterol molar ratio and justify the selected composition

-          Pg 4, Section 2.6. Encapsulation efficiency determination.

The extra-liposomal lysozime was determined by direct mix of liposomes with BCA. With this procedure there is a high risk of leaking and  lysozime inside the liposomes reacting with BCA. For determination of extra-liposoma lysozime, liposomes should be  separated by centrifugation and  the supernatant should be  used for quantification

-          Pg 4, Section 2.7. Modelling of fluid mixing , line 154.

The authors asume that the flow was in the steady steady , laminar régimen. This assumption shoul be justified

-          Pg 6, lines 208- 209 :

“The pdi values of all the produced liposomes varied between 0,15 to 0,25”. Nevertheless,  images of liposomes shown in Figure 3 panel C: FRR 3:1 reveal high size dispersión. ¿How is this explained?

-          Pg 8,

 Figures 4 and 5 show particle size data that do not match particle size data  in Table 1. In the  figures particle size from 100 nm to 600 nm are shown  but from 120.9 ±0.7 nm  to 148.9 ±3.3 nm are reported in Table 1. Please clarify this.

-          Page 10 Section 3.2.2. Erythromycin-loaded liposomes.

Since Erythromycin is incorporated into the lipid bilayer of liposomes it would be interesting to analyze the lipid/drug molar  ratio  and to check whether  the initial PC/colesterol ratio is maintained in the  erythromycin- loaded liposomes, or not.

-          Pg13, section Drug encapsulation efficiency , lines 411-415

Lipid/drug molar ratio should be taken into account for hydrophobic drugs.  This might explain the results of not increase of EE% For erythromycin.  This issue should be adderessed by the authors

-          References

Very few recent papers (only two from 2019, one from 2020 and 1 from 2021) are cited. The topic of this manuscript is of high interest currently and there are several publications from the last two years addressing microfluidics for liposome preparation thar should be commented in the introduction and cited

Reviewer 3 Report

The present work reports the applicability of multi-inlet vortex mixer on the preparation of liposomes with or without encapsulated drug. The tested alternative inlet geometries did not introduced the advantages that may be hypothesised. The paper describes well the techniques performed and presents a new method for liposome preparation.

Some questions/corrections suggested:

- What was the volume prepared for each liposome batch and what was the mean duration of the process to prepare such volume?

- Ln 9: “nano-liposome”: use only liposome as liposomes with drug delivery interest are nanosized structures

- Ln 11: remove comma

- Ln 87: remove colon

- Ln 89: zigzag instead of Zigzag

- Ln143-144: Please, specify how you can ensure the perfect separation of unentrapped drug from drug-loaded liposomes using a 0.4 µm filter. The size of erythromycin vesicles is <200 nm.

- Ln162/177: A “scheme” or a “schematic representation”

- Ln177: End sentence with full stop.

- Ln 207: “under” instead of “on”

- Table 1 caption and first row: pdi instead of PDI

- Ln 263: “core” instead of “cores”
